# Seasonal Variation in Physiological Traits of Amazonian *Coffea canephora* Genotypes in Cultivation Systems with Contrasting Water Availability

Aldo Max Custodio [1,2,*], Paulo Eduardo de Menezes Silva [2], Thiago Rodrigues dos Santos [3],
Lucas Loram Lourenço [2], Roniel Geraldo Avila [2], Anderson Rodrigo da Silva [2],
Fernando Higino de Lima e Silva [2], Marcelo Curitiba Espindula [4], Jairo Rafael Machado Dias [3]
and Fabiano Guimarães Silva [2]

1 Instituto Federal de Educação, Ciência e Tecnologia de Rondônia, Colorado do Oeste 76993-000, Brazil
2 Instituto Federal de Educação, Ciência e Tecnologia Goiano, Rio Verde 75901-970, Brazil
3 Department of Agronomy, Campus Rolim de Moura, Universidade Federal de Rondônia, Rolim de Moura 76940-000, Brazil
4 Empresa Brasileira de Pesquisa Agropecuária, Unidade Rondônia, Porto Velho 76815-800, Brazil
* Correspondence: aldo.custodio@ifro.edu.br

**Abstract:** Climate variation throughout the year affects photosynthesis and other physiological processes correlated with plant development and yield. This study aimed to evaluate the changes in the physiological attributes of *Coffea canephora* genotypes over the year in the Brazilian Amazon and assess their relationship with crop yield. The experiment was carried out in three cultivation systems with three genotypes. The evaluations were carried out in four periods: the peak of the dry season (S1); the beginning of the rainy season (S2); the peak of the rainy season (S3); and the beginning of the dry season (S4). A dataset of gas exchange, pigment indices, chlorophyll fluorescence, branch growth, and coffee yield was obtained. The group of gas exchange variables was the main contributor to treatment discrimination and was most affected by seasons. As expected, the values of $g_s$, $E$, and $A$ were significantly lower in S1, while the values of $VPD_{Leaf-ar}$, $T_{Leaf}$, and IWUE were significantly higher. Our results demonstrate that climatic seasonality affects the photosynthesis of Amazonian Robustas coffee, even under irrigated conditions, particularly in response to increased VPD. The physiological variables analyzed at the leaf level, even in different periods, did not explain the differences in the yield of *C. canephora*.

**Keywords:** coffee tree; Amazonian Robusta; irrigation; photosynthesis; gas exchange; anthocyanin

## 1. Introduction

The annual climatic seasonality of a region is mainly characterized by variations in the incidence of solar radiation, temperature, rainfall, wind, and air relative humidity [1,2]. These variations can affect morphological, physiological, and biochemical processes, including *C*-assimilation and partitioning, respiration, nutrient uptake, translocation, and whole metabolism, increasing or reducing crop productive potential of the *Coffea canephora* Pierre ex Floehner [3–6]. Approximately 15% of the Brazilian production of *C. canephora* is from the southwestern Brazilian Amazon, mainly from the state of Rondônia. This region is characterized by two well-defined seasons throughout the year: a rainy season (October to May), and a dry season (June to September). Both seasons have a high average temperature, small thermal amplitude, and high light availability [7,8].

In the rainy season, the high availability of water in the soil, high relative humidity, and low vapor pressure deficit (VPD) favor the opening of the stomata of coffee trees [9–11]. The increased stomatal opening leads to an increase in stomatal conductance ($g_s$) and, consequently, enhances the diffusion of $CO_2$ into the leaves [12]. The increase in the internal

concentration of $CO_2$ ($c_i$) increases the net photosynthetic rate ($A$). Under these conditions, the risks of irreversible damage to the photosynthetic apparatus ("photoinhibition") are reduced due to the high demand for energy (ATP and NADPH), even at high temperatures and/or light intensity [13]. On the other hand, during the dry season, water restriction can lead to partial or total closure of stomata throughout the day, decreasing $g_s$ and $c_i$, thus compromising $A$ [14]. However, in the biochemical phase of photosynthesis, the decrease in $CO_2$ fixation reduces the energy demand and triggers a cascade effect. This can result in photoinhibition and impairment of the photosynthetic apparatus due to the formation of reactive oxygen species (ROS) [13]. High light intensity and temperature can aggravate this situation [3,15].

Compared to other tropical plant species, coffee has low $A$ values (4 to 11 μmol $CO_{2\,m}{}^{-2}$ s$^{-1}$) [5]. Diffusive barriers seem to be the main photosynthetic limitation of coffee plants [16], their stomata are very sensitive to both soil water restriction and increased VPD [11,17,18]. On the other hand, coffee plants have a series of protection mechanisms against photoinhibition and photodamage, mitigating possible effects of excess radiation associated or not with a reduction in photochemistry [10,19,20]. However, under stress conditions, such as high temperatures and drought, excess radiation can damage photosystems, resulting in leaf scald [15]. In the literature, there are a reasonable number of studies evaluating the isolated effects of an environmental and climatic variable on coffee physiology, however, under cultivation conditions, stress factors occur simultaneously and interact with each other [3,4,11,20]. Thus, the effects on physiology, notably on the photosynthetic process, can be variable according to region, time of year, and genotype [5,16]. In addition, the responses may vary according to the phenological stage [21], although, in this case, part of the differences can be explained by the environmental characteristics that condition the phenology of the plant. Other studies indicate that fruit load also affects the photosynthetic rate of coffee in a positive way [22] and seems to be related to leaf $g_s$ [23,24].

Climate anomalies and increasingly frequent and intense droughts in the Amazon [25] have made irrigation crucial for coffee cultivation in the region. Complementary water supply reduces the risk of damage from intense and/or prolonged water deficits, especially when they occur during the reproductive growth phase [3,4,26]. Greater vigor, growth, and coffee yield are expected in irrigated cultivation systems than in dryland farming systems [27,28]. In the state of Rondônia, there is a predominance of localized irrigation as the management technique in coffee plantations. This irrigation method results in water and energy savings, compared to other methods, and allows for fertigation (aqueous fertilization). Compared with the conventional fertilization system in irrigated crops, fertigation can increase the efficiency of fertilizer use, vegetative growth, yield of coffee trees, and the quality of the coffee beans [29], although this is not always the case [30]. Fertigation usually reduces labor costs, however, recommended fertilizers are more expensive.

Coffee cultivation in irrigation and/or fertigation systems enables high yield in the Amazon region, with an average yield above 6 MT/ha/year of clean (green) coffee [31,32]. However, the additional water supply to trees through irrigation may not be enough to avoid the negative effects of drought on coffee photosynthesis [12]. In this case, VPD is a key variable in inducing stomatal opening and closing. High VPD conditions lead to a considerable increase in tree transpiration rates [33,34], and can lead to significant drops in coffee yield [35]. In some situations, it can even overcome the capacity of the tree hydraulic system, which is inherent to the genotype, to absorb and/or transport water from the soil to the leaves, even with water available in the soil [12,24].

The main coffee genotypes grown in Rondônia are the result of the natural hybridization of genotypes belonging to the Group SG1 or "Conilon" [36,37] from the state of Espírito Santo on the eastern Brazilian coast, with other genotypes of an undefined genetic group phenotypically described as "robusta" [31,38], which were introduced from the Germplasm Bank of the Agronomy Institute of Campinas (IAC), Campinas, Brazil. These hybrids are locally known as Robustas Amazônicos, or Robustas of Amazônia (Amazonian Robustas) [31] and stand out for their high vigor, tolerance to diseases [18], high yield [18,19], and

superior drinking quality [38]. However, these genotypes are generally more sensitive to drought than "Conilon" [31]. In recent years, several genotypes of Amazonian Robustas have spread to other producing regions of *C. canephora* in Brazil, mainly in the state of Espírito Santo.

Despite the success of coffee production in Rondônia, little information is available on the ecophysiology of coffee trees grown in the Brazilian Amazon. In addition, most of the studies on the effect of environmental and climatic variables on coffee metabolism were limited to analyses of processes that occur in the leaves, mainly gas exchange, chlorophyll *a* fluorescence, and hydraulic and antioxidant metabolism [5,16,39,40]. These assessments allow us to understand how variable and stress conditions can affect different stages of photosynthesis in the plant. However, environmental conditions with a low impact on photosynthesis can result in significant reductions in coffee productivity [4]. Therefore, in some situations, photosynthesis-related assessments, such as gas exchange, may not have a direct relationship with yield [4]. Therefore, more studies are needed to associate the common assessments made on the leaves with the yield of coffee trees. The aim of this study was to evaluate the seasonal variation in physiological variables related to photosynthesis in genotypes of Amazonian Robustas under fertigation, irrigation, and dryland farming, cultivated under field conditions in the Amazon, and to correlate these variables with yield.

## 2. Materials and Methods

### 2.1. Experimental Area and Coffee Cultivation

The research was carried out at the experimental farm of the Federal University of Rondônia Foundation (*Fundação Universidade Federal de Rondônia—UNIR*), located in the municipality of Rolim de Moura, Rondônia state, Brazil (11°34′5″ S; 61°41′12″ W; altitude 275 m). The region's climate is of type Aw (tropical with dry winter), according to the Köppen classification [8]. The experimental area has a flat topography and dystrophic red-yellow latosol (ferralsols) soil [41]. Precipitation and temperature data of the region were obtained from an automatic station for the study period (July 2019–July 2020) (Figure 1) from the Meteorological Institute (Instituto Nacional de Meteorologia–INMET) of Brazil [42]. On the evaluation days, the average values of maximum, average, and minimum temperature, and average and minimum relative humidity were obtained (Table 1). From these data, the average and maximum air vapor pressure deficits (VPD) were calculated, using the difference between the vapor saturation pressure (es) and the partial vapor pressure (ea) [43].

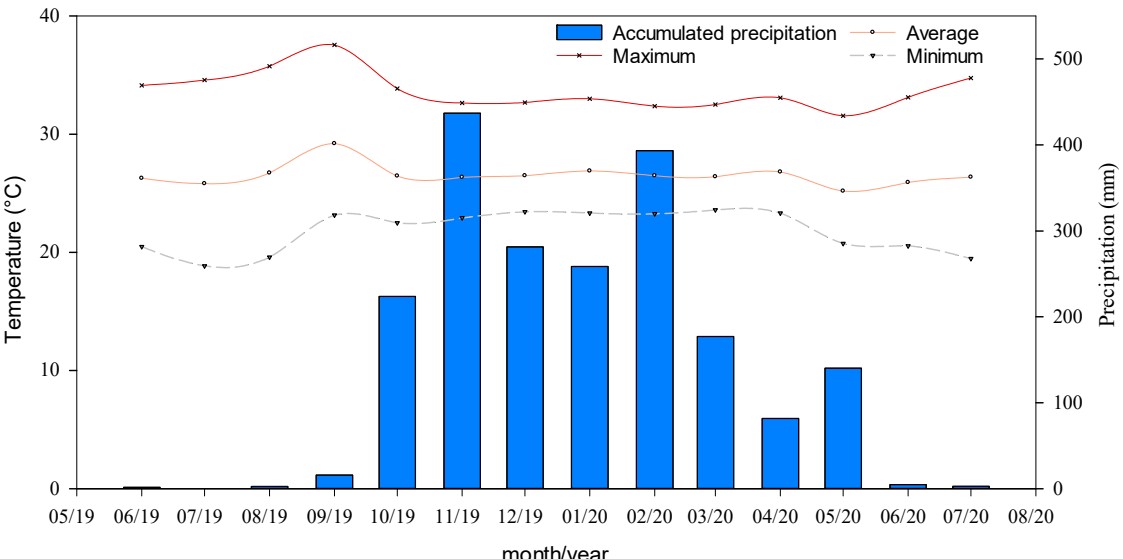

**Figure 1.** Accumulated precipitation data (blue) and minimum (grey), average (orange) and maximum (red) mean temperatures of the region during the experimental period.

**Table 1.** Climatic data of the region during the assessments. Average of daily mean values.

| Season | Rel. Air Humidity (%) | | Temperature (°C) | | | VPD (kPa) | |
|--------|---------|---------|---------|---------|---------|---------|---------|
| | Minimum | Average | Minimum | Average | Maximum | Average | Maximum |
| S1 | 30.0 | 56.5 | 22.5 | 28.7 | 36.4 | 1.38 | 3.31 |
| S2 | 50.3 | 78.7 | 22.7 | 26.3 | 32.7 | 0.59 | 1.93 |
| S3 | 60.7 | 83.9 | 23.1 | 25.8 | 30.9 | 0.44 | 1.39 |
| S4 | 38.0 | 69.0 | 20.2 | 25.8 | 33.1 | 0.84 | 2.46 |

The coffee plantation was established in November 2013, with seedlings of Amazonian Robustas (*Coffea canephora* Pierre ex Floehner) of clonal origin spaced at 3.0 m between rows and 1.5 m between trees. Part of the area was cultivated with supplementary irrigation (fertigation and irrigation systems) and part without irrigation (dryland farming). The average yield of the three previous harvests was 7.99 MT/ha under fertigation (2018/2019), 7.86 MT/ha under irrigation (2017/2018), and 6.40 MT/ha under dryland farming (2016/2017) of green coffee, with no difference between cultivated genotypes. Preliminary analysis showed statistical differences only for the dryland farming system between the 2016/2017 crop and crops of the other two years (unpublished data).

Fertilization of the coffee plantation was carried out according to [44], aiming for a yield of 6.0 to 7.8 MT/ha of green coffee. In the irrigation and dryland farming systems, the distribution of fertilizers was conventional (manual) in the projection of the coffee canopy area. In the fertigation system, fertilization was carried out using a localized irrigation system after the solubilization of fertilizers. Cultivation treatments were carried out throughout the experimental area according to [45] and the technical recommendations for the Rondônia region [46]. In the fertigation and irrigation cultivation systems, irrigation was carried out with drip systems in June, July, August, and September of 2019. However, irrigation was suspended between 15 July and 10 August 2019. This procedure is common in production units in the region and aims to increase uniformity in flowering and, consequently, in fruit ripening.

*2.2. Experimental Design*

The experimental area was divided into three separate cropping systems [irrigation (I), fertigation (F), and dryland farming (D)], but in adjacent areas. In each cultivation system, the experiment was arranged in split-plot a randomized block design (RBD) with four replicates. Three genotypes [clones "25" (G1), "08" (G2), and "03" (G3)] were evaluated in four seasons (S1, S2, S3, and S4). The three genotypes are highly productive and the most cultivated in Rondônia [32,38,47]. No studies detailing the morphological, physiological, and/or genetic aspects of these genotypes have been published to date. According to field observations, clone "03" differs mainly from the other two genotypes by the opening of its crown, resulting from the inclination of its stems, mainly due to the high production. Scalded leaves are common in this material during the dry season, probably due to photoinhibition. Clone "25" has leaves with a larger area and normally suffers from water restriction more quickly than the other genotypes. Clone "08" shows good vigor and initial development but has a high rate of natural fruit abortion in the initial stages. The three genotypes were selected by the same coffee producer of the study region and the first clonal seedlings were produced in the 1999/2000 harvest.

The evaluation periods were selected to assess climatic seasonality effects on crops. The four seasons studied were as follows:

1. S1—peak of the dry season, with evaluations carried out between 23 and 25 August 2019. During this period, trees were in pre-anthesis (Biologische Bundesanstalt, Bundessortenamt and Chemical Industry scale—BBCH—58,59 [48]) in the dryland farming system and at the beginning of anthesis (BBCH 61–65) in the irrigation and fertigation systems. In this season there is a reduced rate of vegetative growth.

2.    S2—beginning of the rainy season, with evaluations carried out between 21 and 24 November 2019. During this period, the accelerated growth of the fruit occurs (BBCH 73–75) and the resumption of the vegetative growth rates.
3.    S3—peak of the rainy season, with evaluations carried out between 28 February and 1 March 2020. During this period, the final stage of bean filling occurred (BBCH 77–79). At this season there are high rates of vegetative growth.
4.    S4—beginning of the dry season, with evaluations carried out between 26 and 28 June 2020. In this period, trees had been harvested 20 days before and, atypically, had only partially flowered (BBCH 58–65) (2020/2021 harvest). In this season vegetative growth is stopped (dryland farming system) or reduced (irrigation and fertigation systems).

### 2.3. Assessment of Physiological Variables Associated with Photosynthesis

Evaluations of pigment and chlorophyll a (*Chl a*) and b (*Chl b*) indices, *Chl a* fluorescence, and gas exchange were performed out in the morning on fully expanded leaves of the third or fourth nodes from the apex of the plagiotropic branch, chosen from the upper third of the tree. The analyses were carried out in two plants i.e., two leaves of each experimental unit, whose mean was used for statistical analyses. The specific procedures for the assessment of each variable group are described below.

### 2.3.1. Leaf Pigment Index

Chlorophyll, flavonoid, and anthocyanin indices were determined using an instantaneous, non-destructive chlorophyll and polyphenol meter from DUALEX Scientific[TM] (Force-A, Orsay, France) [49]. The nitrogen balance index (NBI) was calculated as the ratio between the chlorophyll index and the flavonoid index. For S2, the data for these variables were obtained indirectly through statistical imputation using multivariate regression.

In addition, the chlorophyll index was also determined using a Chlorofilog CFL-1030 FALKER chlorophyll meter (Automação Agrícola, Porto Alegre, Brazil) to evaluate theindices of *Chl a* and *Chl b* separately. The ratio of *Chl a/Chl b* and total chlorophyll (*Chlt = Chl* a + *Chl b*) were calculated.

### 2.3.2. Chlorophyll a Fluorescence

The transient OJIP fluorescence of *Chl a* was measured using a portable FluorPen FP100 fluorometer (Photon Systems Instruments, Drasov, Czech Republic) following the manufacturer's instructions. The leaves were adapted to the dark for 30 min and then exposed to a saturating pulse of blue light (3000 µmol m$^{-2}$ s$^{-1}$). Data were obtained from fluorescence intensity at points $O$ ($F_o = F_{50\mu s}$), J ($F_j = F_{2ms}$), I ($F_i = F_{60ms}$), and P ($F_p = F_m$) and from the fluorescence intensity at =300 µs ($F_{300\mu s}$). From the fluorescence data, the maximum variable fluorescence ($F_v = Fm - Fo$), the ratios $F_m/F_o$ and $F_v/F_o$, the proportion of F$v$ until $F_j$ ($V_j$) and until $F_i$ ($V_i$), and the parameters of the *JIP*-test were calculated according to previous reports [50,51]. Specific flow parameters were obtained by primary quinone reduced ($Q_a{}^-$) by reaction center (*RC*) (t = 0), and by the absorbed (*ABS/RC*), trapped (*TR$_o$/RC*), dissipated (*DI$_o$/RC*), and transported (*ET$_o$/RC*) energy. In addition, we calculated the maximum (*Phi$_{Po}$*) and average (*Phi$_{pav}$*) quantum yields of the primary photochemical reactions of photosystem II (PSII) (t = 0), the quantum yield of energy dissipation (*Phi$_{Do}$*) and electron transport (*Phi$_{Eo}$*), and the probability of an exciton moving an electron in the electron transport chain beyond $Q_{a.}{}^-$ (*Psi$_o$*). We also obtained the area under the fluorescence curve (Area), the area under the fluorescence curve between $F_{40\mu s}$ and $F_{1s}$ (FixArea), normalized area ($S_m$), smallest possible $S_m$ ($S_s$), approximate initial slope of the fluorescence transient normalized with $F_V$ ($M_o$), the turnover number of $Q_a$ (N), ratio of reaction centers and the absorbance (*RC/ABS*), and the performance index on the absorption base (*PI$_{ABS}$*).

### 2.3.3. Gas Exchange

Using a portable infrared gas measurement system (IRGA; LI-COR—Li 6800), we obtained the following traits: the net assimilation rate of $CO_2$ ($A$; $\mu$mol $CO_2$ m$^{-2}$ s$^{-1}$); transpiration rate ($E$; mmol H$_2$O m$^{-2}$s$^{-1}$); ambient (external) $CO_2$ concentration ($Ca$; $\mu$mol/mol); internal $CO_2$ concentration ($Ci$–$\mu$mol/mol); stomatal conductance ($g_s$—mol m$^{-2}$s$^{-1}$); VPD between the leaf and the air of the IRGA chamber (VPD$_L$; kPa); and leaf temperature (T$_L$—$^\circ$C). The evaluations were carried out with an ambient concentration of 400 $\mu$L L$^{-1}$ $CO_2$ and an irradiance of 1000 $\mu$mol m$^{-2}$ s$^{-1}$. From the equipment data, we calculated the instant carboxylation rate ($A/Ci$), $Ci/Ca$ ratio, instant water use efficiency (WUE = $A/E$), and intrinsic water use efficiency (iWUE = $A/g_s$).

### 2.3.4. Relative Water Content (RWC) in Leaves

To determine the RWC, four segments measuring 1 cm$^2$ were cut from six fresh leaves per experimental unit and weighed on a precision balance to verify fresh mass (FM). The segments were then immersed in distilled water for 24 h to obtain turgid mass (TM). Subsequently, the plant material was placed in an oven at 70 $^\circ$C for 24 h to obtain the dry mass (DM). RWC was calculated according to Equation (1) and only during S1, when the coffee trees were exposed to different conditions of water supply and deficit, according to the treatments.

$$\text{RWC} = \frac{\text{FM} - \text{DM}}{\text{TM} - \text{DM}} \tag{1}$$

### 2.4. Vegetative Growth and Production

One vegetative (orthotropic) and two reproductive (plagiotropic) branches were marked on the trees at the beginning of the production cycle and their length was measured monthly between August 2019 and May 2020. From the data obtained, the average daily growth rate (cm/day) was calculated. In the first week of June 2020, all fruits were harvested from four trees in each experimental unit. We then estimated the average yield (MT/ha) of dried cherry coffee and green coffee.

### 2.5. Statistical Analysis

The data of the physiological variables (gas exchange, pigment index, and fluorescence) were subjected to multivariate analysis of variance (MANOVA) according to the joint analysis model (by cultivation system) of experiments in plots (by clone) split in time (by season), using randomized blocks. The residuals were tested for multivariate normality using the generalized Shapiro—Wilk test [52], and for homogeneity of covariance matrices using the Box's M test. The Box–Cox [53] procedure was used to avoid variables escaping both assumptions. The bean yield data were subjected to joint analysis of variance and Fisher's LSD test to compare the means by clones and systems. From the MANOVA performed for the physiological variables, canonical discriminating variables were built to compare the factor levels using ellipses of 95% confidence. The scores for the first two canonical variables were represented in a biplot [54]. The residual correlation matrix was graphically represented in a correlation network, as described by [55]. We attempted to identify the correlation patterns of groups of physiological variables with bean yield. Additionally, a correlation analysis was carried out between the mean values of VPD, RH, and ambient temperature (Ta), and climatic variables of the assessment periods in each season with the mean values of the gas exchange variables ($g_s$, $c_i$, $VPD_L$, and T$_L$). First-order autoregressive models were fitted to the time series of the plagiotropic and orthotropic branches to estimate average growth (p.med and o.med, in cm/day) and the time dependency parameter plagiotropic and orthotropic branches (p.ar1, o.ar1). These estimates were subjected to canonical discriminant analysis, as described above, for the comparison of clones and systems.

The analyses were performed in R (version 3.5.3), using the packages "biotools" [56] "candisc" [57], and "qgraph" [58].

## 3. Results

August and September 2019 were characterized by high temperatures, absence of rain (Figure 1), and low RH. In the dry season, RH reached values equal to or less than 20% during at least 13 days. In September, the maximum VPD was estimated to reach above 3.5 kPa, while the average value was around 1.5 kPa. Starting in October 2019, a small decrease in temperature and an increase in RH was verified with a significant accumulation of rain. High precipitation (437 mm) was recorded in November 2019, resulting in high RH, which exhibited little fluctuation throughout the rainy season until May 2020. There was no rainfall in the region in June 2020, resulting in decreased RH. VPD decreased with the increase in RH in October and remained low with small oscillations until May 2020, when it rose again until the experiment was completed at the end of June (Table 1).

The first two canonical discriminating variables, canonical 1 (Can. 1) and canonical 2 (Can. 2), constructed using a MANOVA for physiological variables, retained 75% of the multivariate differences between systems, clones, and periods, with 64.7% in Can. 1 (Figure 2). The group of variables related to gas exchange (mainly $VPD_L$, $T_L$, IWUE, $c_i$, $g_s$, $E$, and $A$) highly contributed to the discrimination of treatments. A high coefficient was also observed for the anthocyanin index. The variables related to the chlorophyll index had low relative discrimination power. Chlorophyll fluorescence parameters contributed more to the second canonical discriminating variable (Can. 2), together with $A$ and $A/Ci$ (Figure 2), than other variables.

We found a significant effect ($p < 0.05$) of the evaluation period on the physiological variables, in addition to a trend concerning Can. 1 (Figure 2). This effect was superior to that of the system and clone factors (Figure 3). The most contrasting periods were S1 (drought peak) and S3 (rainy peak). In S1, we observed higher values of anthocyanin index, $T_L$, and $VPD_L$ than those in other periods. In S3 and S4, we detected higher values of $g_s$, $c_i$, $E$, and $A$ than those in other periods. In the latter two seasons, systems and clones caused less dispersion in physiological responses (Figure 2). In S1, we found a significant difference between the dryland farming system ($p < 0.05$) and the fertigation and irrigation systems, considering all clones. Clone "03" differed from clones "25" and "08" when cultivated in the dryland farming system, especially regarding chlorophyll *a* fluorescence and gas exchanges traits (Figure 2).

We observed stronger correlations between variables in the same group of physiological analyses (gas exchange, chlorophyll *a* fluorescence, and pigments indexes) than in those between the groups (Figure 4A). There was a positive correlation between $g_s$ and $c_i$ and between both variables and $A$ ($p < 0.01$). In contrast, $g_s$ was negatively correlated with $VPD_L$ and $T_L$ ($p < 0.01$). The values of $g_s$ during the evaluation periods were also negatively correlated with VPD ($p < 0.1$) and positively correlated with RH ($p < 0.1$) (Table 2). The variable $c_i$ was negatively correlated with environment temperature ($T_e$) ($p < 0.05$). For gas exchange variables, the correlations were high within the seasons, whereas the correlations between seasons were low (Figure 4B). We found a correlation between the dried cherry coffee yield and the gas exchange variables in S2, with absolute values around 0.60. In the other periods, these correlations were below 0.15, considering absolute values (Figure 4B).

**Table 2.** Correlation of environmental variables (temperature—Te, relative humidity—RH, air vapor pressure deficit—VPD) with gas exchange variables (stomatal conductance—$g_s$, internal $CO_2$ concentration—$C_i$, leaf temperature—$T_L$, vapor pressure deficit between the leaf and the air of the IRGA chamber—$VPD_L$).

| Environmental Variables | Gas Exchange Variables | | | |
|---|---|---|---|---|
| | $g_s$ | $Ci$ | $T_L$ | $VPD_L$ |
| Te | −0.78 | −0.98 * | 0.99 * | 0.96 * |
| RH | 0.94 [+] | 0.92 [+] | −0.94 [+] | −0.94 [+] |
| VPD | −0.90 [+] | −0.93 [+] | 0.97 * | 0.97 * |

+ significant with $p < 0.1$; * significant with $p < 0.05$.

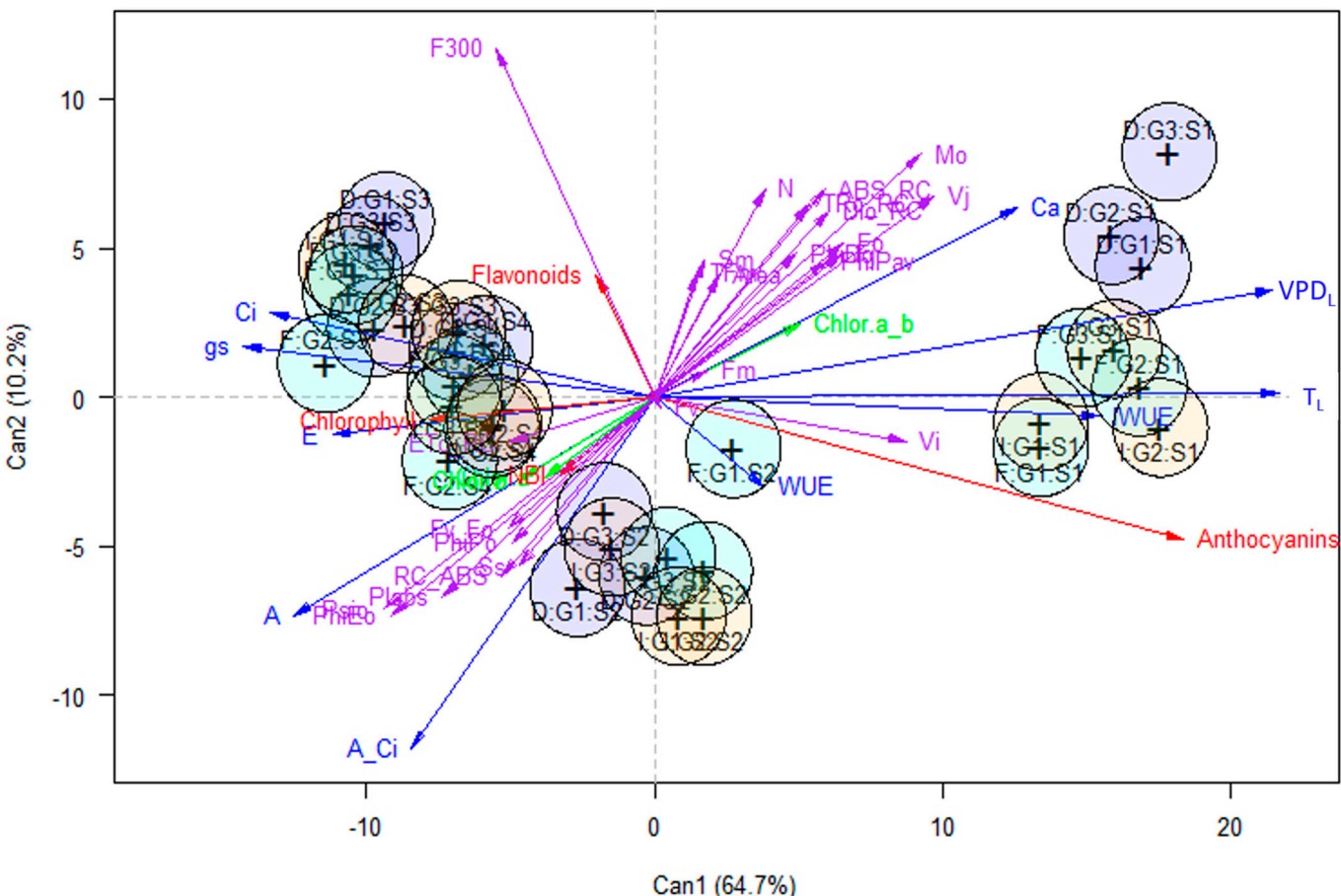

**Figure 2.** Biplot containing mean scores of the first two canonical (Can. 1 and Can. 2) discriminating variables constructed from gas exchange variables, chlorophyll and pigment indices, and chlorophyll *a* fluorescence trait. The 95% confidence ellipses for the combinations of system, clone, and period are shaded. (Systems: F—fertigation, I—irrigation, D—dryland farming; genotypes: G1—clone "25", G2—clone "08", G3—clone "03"; Season: S1—peak of the dry season, S2—beginning of the rainy season, S3—peak of the rainy season, S4—beginning of the dry season). A—net assimilation of $CO_2$, E—transpiration, Ca—ambient (external) $CO_2$ concentration, Ci—internal $CO_2$ concentration, gs—stomatal conductance, VPD—vapor pressure deficit between the leaf and the air of the IRGA chamber, Tl—leaf temperature, A_Ci—*A/Ci*, WUE—instant water use efficiency, IWUE—intrinsic water use efficiency. Chlor.a—chlorophyll *a* index, Chlor.b—Chlorophyll *b* index, Chlor.a_b—ratio of *Chl a/Chl b*, NBI—nitrogen balance index, Fo—initial fluorescence (t = $_{50\mu s}$), Fm—maximum fluorescence, F300—fluorescence intensity at =300 μs, TFm—Fm time, Fv—variable fluorescence, Fv_Fo—$F_v/F_o$, Vj—the proportion of F$v$ until $F_j$ (*j*), Vi—the proportion of F$v$ until $F_i$ (*i*), *ABS*_RC-absorbed, TRo_RC—trapped, DIo_RC– dissipated and ETo_RC—transported energy by primary quinone reduced ($Q_a^-$) by reaction center at t = 0, PhiPo—maximum and PhiPav—effective quantum yields of the primary photochemical reactions of photosystem II (PSII) (t = 0), PhiDo—quantum yield of energy dissipation, PhiEo—quantum yield of electron transport, Psio—probability of an exciton moving an electron in the electron transport chain beyond $Q_{a.}^-$. Area—area under the fluorescence curve, Sm—normalized area, Ss—smallest possible $S_m$, Mo—approximate initial slope of the fluorescence transient normalized with $F_V$, N—the turnover number of $Q_a$, PIabs—performance index on the absorption base.

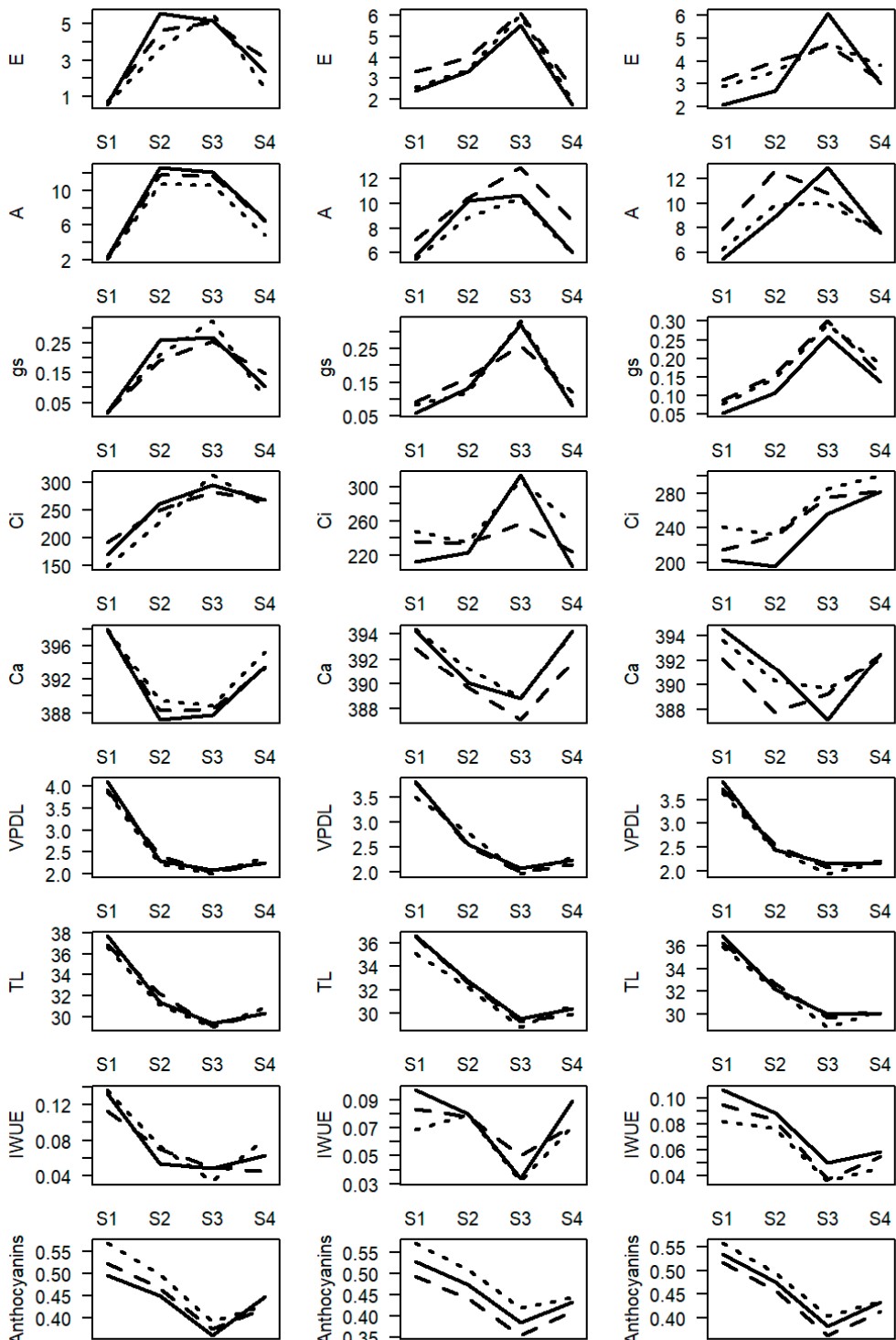

**Figure 3.** Averages of the physiological variables of coffee with the greatest discriminating power between systems (dryland farming in the left column, irrigation in the center, fertigation in the right), genotypes (clone "25", continuous line; clone "08", dashed line; clone "03", dotted line), and seasons (S1—peak of the dry season; S2—beginning of the rainy season; S3—peak of the rainy season; S4—beginning of the dry season). E—transpiration rate (mmol $H_2O$ m$^{-2}$s$^{-1}$), A—net assimilation rate of $CO_2$ (μmol $CO_2$ m$^{-2}$ s$^{-1}$), gs—stomatal conductance (mol m$^{-2}$s$^{-1}$), Ci—internal $CO_2$ concentration (μmol/mol), Ca—ambient (external) $CO_2$ concentration (μmol/mol), VPD—vapor pressure deficit between the leaf and the air of the IRGA chamber (kPa); Tl—leaf temperature (°C). IWUE—intrinsic water use efficiency ($A/g_s$).

In S1, we found a statistical effect only of the cultivation system when considering the effect of clones and cultivation systems on the RWC of leaves (Figure 5). The RWC was higher in the irrigation and fertigation systems at the three evaluated times than in the dryland farming system. In the predawn, the RWC values in the dryland farming system were, on average, approximately 35% lower than those observed for irrigation systems (Figure 5).

The first canonical discriminating variable retained 91.7% and 98.1% of the multivariate differences between systems and clones, respectively, when considering the average growth rate of stems and branches (Figure 6). The dryland farming system showed the lowest average growth rate of plagiotropic branches, and clone "25" showed the lowest growth rate of the orthotropic stem. We found a high temporal dependence on the growth rate of stems and branches in the fertigation system (Figure 6).

The coffee yield in dried cherry and green coffee showed similar behavior, with the lowest averages observed in the dryland farming system for all clones (Table 3). Clones "08" and "03" had higher yields in the fertigation system than clone "25". We found no difference in the fertigation and irrigation systems for clone "25". The "03" clone had a higher yield than the other two clones in the fertigation and dryland farming systems. In the dryland farming system, its yield was about three and seven times higher than the yields of the "25" and "08" clones, respectively. For the variable Can. 1, we found no difference between cultivation systems concerning clone "03". The fertigation system showed the lowest values for the "08" and "25" clones. In the irrigation and dryland farming systems, clone "08" showed the highest Can. 1. Finally, clone "25" had the lowest values in all cultivation systems.

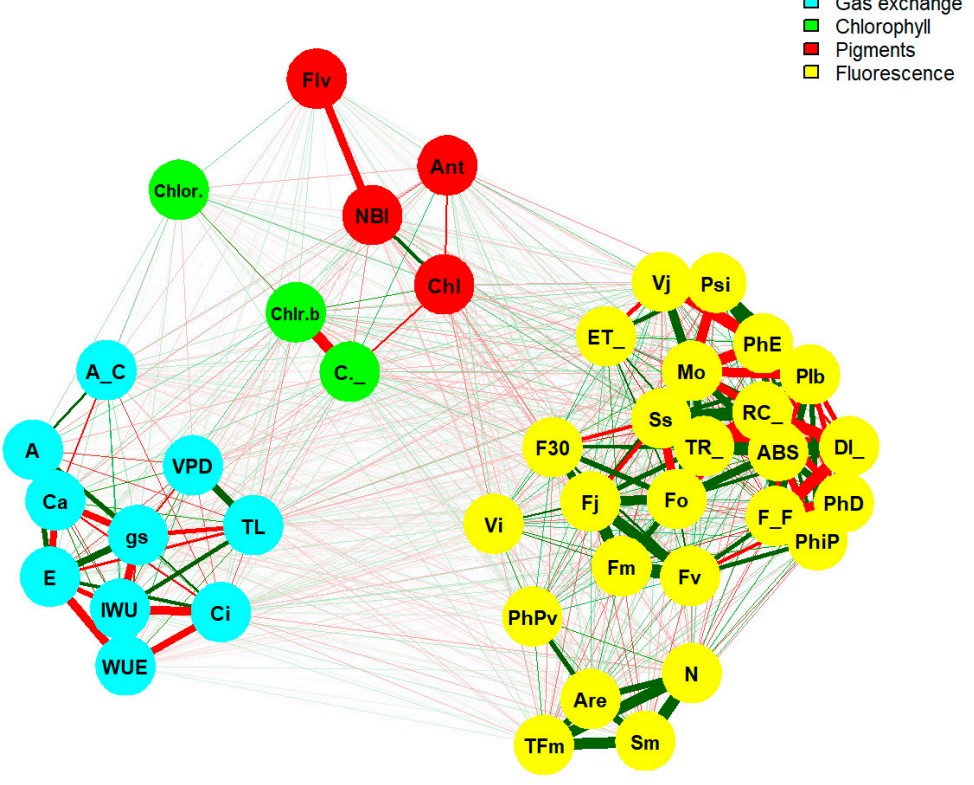

(**A**)

**Figure 4.** *Cont.*

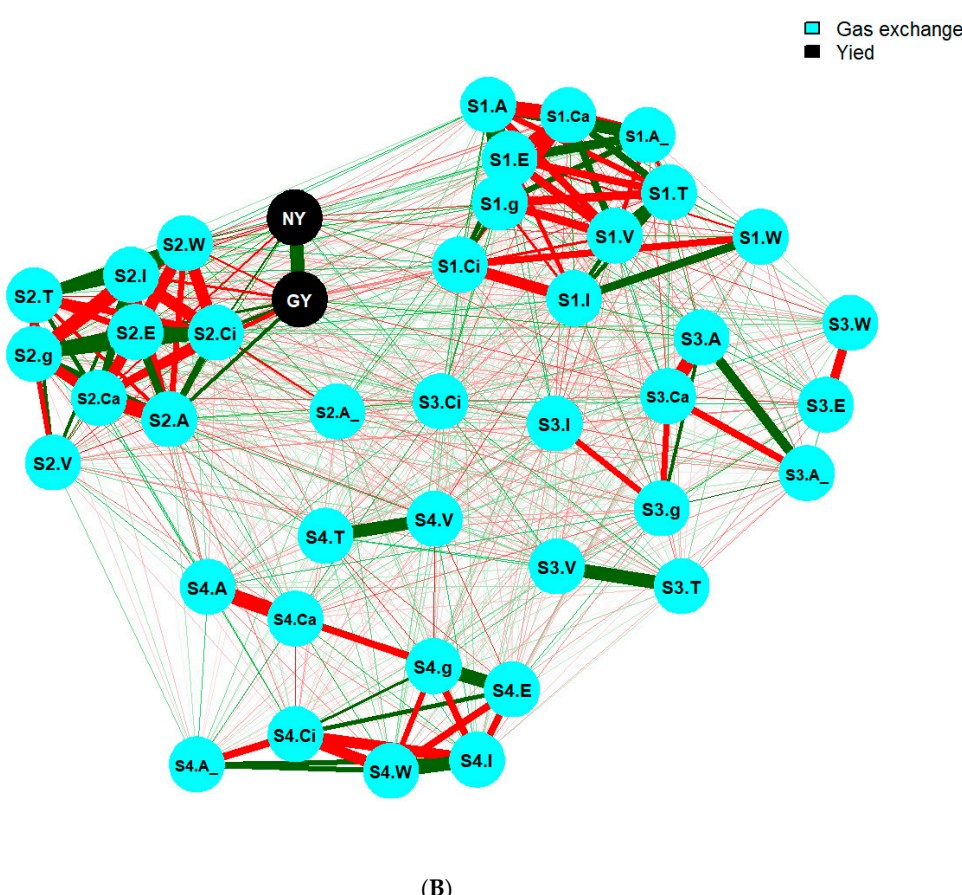

**(B)**

**Figure 4.** (**A**) Correlation network for coffee physiological variables. Correlations greater than 0.5, positive (green line) or negative (red line), are highlighted (thicker lines indicate higher correlations). A—net assimilation of $CO_2$, E—transpiration, Ca—ambient (external) $CO_2$ concentration, Ci—internal $CO_2$ concentration, gs—stomatal conductance, VPD—vapor pressure deficit between the leaf and the air of the IRGA chamber, Tl—leaf temperature, A_C—*A/Ci*, WUE—instant water use efficiency, IWU—intrinsic water use efficiency. Chlor.a—chlorophyll *a* index, Chlor.b—Chlorophyll *b* index, C._—ratio of *Chl a/Chl b*, NBI—nitrogen balance index, Chl—chlorophyll index by Dualex, Flv—flavonoid index, Ant—anthocyanin index, Fo—initial fluorescence (t = $_{50\mu s}$), Fm—maximum fluorescence, F30—fluorescence intensity at =300 µs, Fj—fluorescence in 2 ms, TFm—Fm time, Fv—variable fluorescence, F_F—$F_v/F_o$, Vj—the proportion of F*v* until $F_j$ (*j*), Vi—the proportion of F*v* until $F_i$, *ABS*—absorbed, TR_—trapped, DI_—dissipated and ET_—transported energy by primary quinone reduced ($Q_a{}^-$) by reaction center at t = 0, PhiP—maximum and PhPv—effective quantum yields of the primary photochemical reactions of photosystem II (PSII) (t = 0), PhD—quantum yield of energy dissipation, PhE—quantum yield electron transport, Psi—probability of an exciton moving an electron in the electron transport chain beyond $Q_a{}^-$. Are—area under the fluorescence curve, Sm—normalized area, Ss—smallest possible $S_m$, Mo—approximate initial slope of the fluorescence transient normalized with $F_V$, N—the turnover number of $Q_a$, RC_—ratio of reaction centers and the absorbance, PIb—performance index on the absorption base. (**B**) Analysis by season (S1—peak of the dry season; S2—beginning of the rainy season; S3—peak of the rainy season; S4—beginning of the dry season) was performed concerning the correlation of gas exchange variables and dried cherry coffee (GY) and green coffee (NY) bean yield. A—net assimilation of $CO_2$, E—transpiration, Ca—ambient (external) $CO_2$ concentration, Ci—internal $CO_2$ concentration, g—stomatal conductance, V—vapor pressure deficit between the leaf and the air of the IRGA chamber, T—leaf temperature, A_—*A/Ci*, W—instant water use efficiency, I—intrinsic water use efficiency.

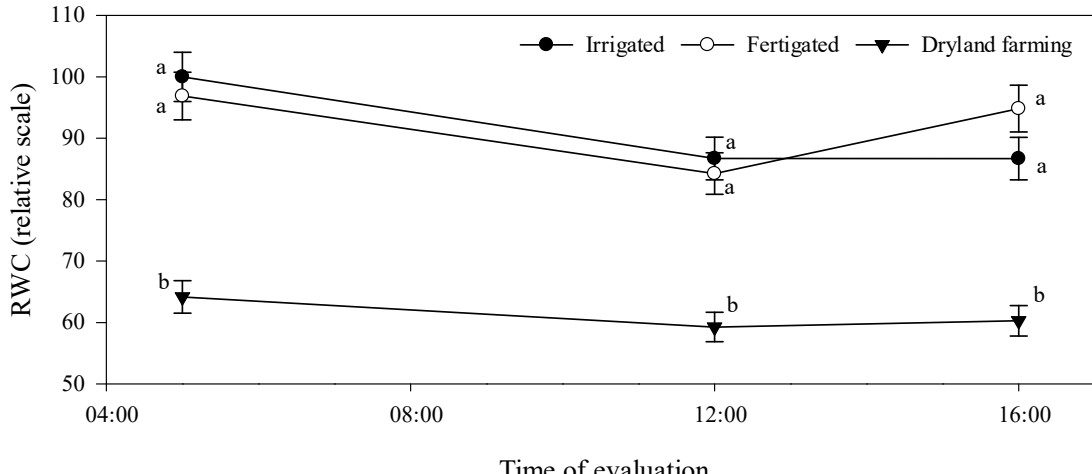

**Figure 5.** Relative water content (RWC) in the leaves of Amazonian Robustas at the time of the peak dry season (S1) in the dawn (05:00), at midday (12:00), and in the late afternoon (16:00). Cultivation systems (irrigation, fertigation, and dryland farming) with means followed by the same letter do not differ by Fisher's LSD test ($p < 0.05$). Bars indicate the standard error of the mean.

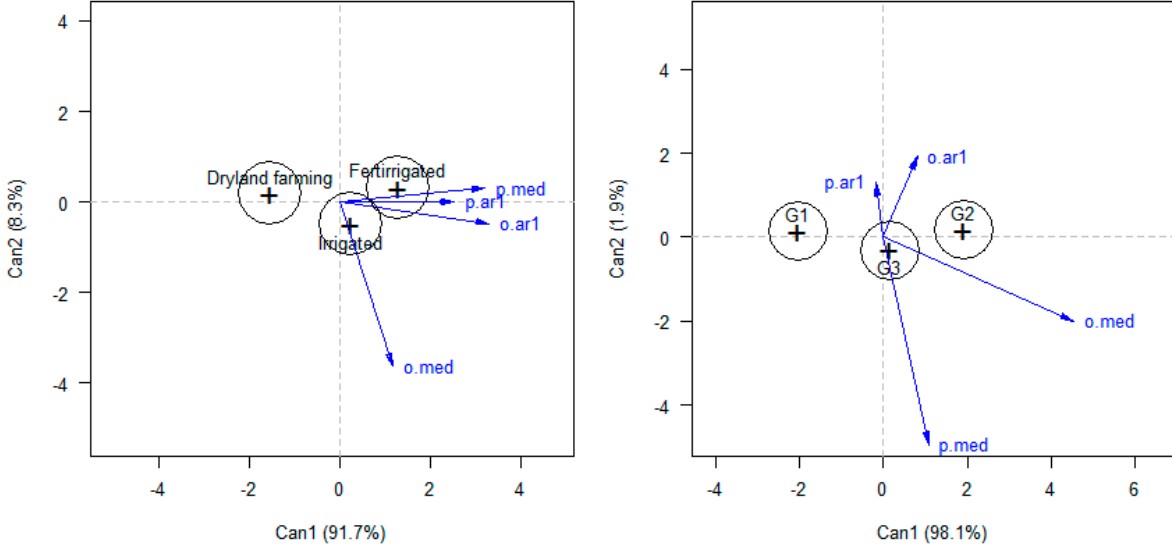

**Figure 6.** Biplot containing mean scores and 95% confidence ellipses for the first two canonical discriminating variables of systems (**left**) and clones (**right**), built from growth variables of plagiotropic and orthotropic branches (p.med and o.med– estimates of the average growth of the plagiotropic and orthotropic groups, respectively; p.ar1 and o.ar1—parameters of temporal dependence on the growth of plagiotropic and orthotropic branches, respectively).

**Table 3.** Average values of production systems and coffee genotypes (G1—clone "25"; G2—clone "08" and G3—clone "03") for the canonical discriminant 1 (64.7%) and bean yield variables.

| | **Can. 1** | | | **Green Coffee (MT/ha)** | | | **Dried Cherry Coffee (MT/ha)** | | |
| | G1 | G2 | G3 | G1 | G2 | G3 | G1 | G2 | G3 |
|---|---|---|---|---|---|---|---|---|---|
| Fertigation | −2.44 bB | 0.56 bA | −0.13 aA | 1.72 aC | 2.53 aB | 4.38 aA | 4.08 aB | 5.37 aB | 8.70 aA |
| Irrigation | −0.83 aC | 1.44 aA | 0.63 aB | 1.82 aA | 1.51 bA | 2.20 bA | 3.77 aA | 3.12 bA | 4.57 bA |
| Dryland farming | −1.06 aC | 1.36 aA | 0.46 aB | 0.45 bB | 0.17 cB | 1.29 cA | 0.94 bB | 0.39 cB | 2.69 cA |
| SVC | − | | | 29.75% | | | 26.14% | | |

Averages followed by the same lowercase letter in the column and uppercase letter in the row do not differ according to Fisher's LSD test at 5% significance.

It is important to highlight that at the end of August and during September, sharp defoliation (abscission) was observed in coffee trees, notably in dryland farming systems, where this phenomenon was visually greater than that commonly observed in the region. In addition, in October, we observed "scalded" and severely damaged leaves in the dryland farming system (field observations).

## 4. Discussion

The long dry season period (especially between August and early October) of approximately 150 days without significant accumulation of rainfall, associated with high temperatures (Figure 1) and high VPD values, imposed unfavorable conditions on the development and production of coffee trees (*C. canephora*) [5,26,35]. The time of the first assessment (S1) showed climatic characteristics and hydric deficit (HD) consistent with those of the drought peak registered for the region in previous years. However, the dry season, which usually ends at the beginning of September, was atypically extended until October. In fact, September 2019 was the driest month in terms of accumulated precipitation and VPD, and was the warmest since 2016. On the other hand, in November, the heavy rain resulted in accumulated precipitation above the monthly average recorded in the previous 11 years of approximately 220 mm [42].

In the dry season (S1), supplementary irrigation improved the indicators of chlorophyll *a* fluorescence, regardless of genotype. It is important to note that with the persistence of the dry period and high temperatures throughout September, "scalded" leaves of coffee trees under the dryland farming cultivation were observed, which were absent during the evaluations at the end of August (S1). Clone "03" showed the lowest values of fluorescence indicators in the dryland farming system, with effects on fluorescence dynamics ($M_o$, $V_j$), quantum efficiency of PSII ($Phi_{Po}$, $Phi_{Eo}$, $Psi_o$, $Phi_{Do}$), and performance index ($PI_{ABS}$) These findings suggest that this genotype may be more sensitive to photoinhibition [13,59].

The behavior of gas exchange traits reflected a cascade effect, triggered by tree responses to temperature seasonality, RH, and VPD (Tables 1 and 2). Low values of $g_s$ were observed at the drought peak, a parameter directly related to the stomatal opening. The $g_s$ vary especially in response to changes in environmental conditions, notably, soil moisture, temperature, and VPD [14,19,59]. The positive correlations between $g_s$ and $E$ ($p < 0.01$), $c_i$ ($p < 0.01$), and $A$ ($p < 0.01$), and negative correlations with $T_L$ ($p < 0.01$), such as those found in this study (Figure 4 A,B), were expected. Stomatal closure is a crucial and efficient strategy to limit water loss from the leaf to the atmosphere, as it reduces $E$. However, this process can result in the reduction of $c_i$ and, consequently, of $A$ [59]. Diffusive factors commonly impose the main photosynthetic limitations on coffee [19,39]. Reduced $E$, in turn, can increase $T_L$, especially on days of high irradiance and high temperatures [14,15].

In the dryland farming systems, the water deficit worsened the effects of climatic seasonality, resulting in lower values of $g_s$, $c_i$, and $A$ than those observed in the systems that received supplementary irrigation. The RWC values showed a HD in trees that were not irrigated since dawn. These results corroborate those of previous studies suggesting that the combination of HD with high temperatures [9,20] and low RH or high VPD reduces $A$ in coffee plants due to diffusive limitations [10,11]. Thus, coffee trees grown in dryland farming systems are more vulnerable to droughts because these events lead to HD associated with high temperatures and/or high VPD in the field [25]. The correlation of $g_s$ and $c_i$ with the climatic variables VPD, RH, and Ta (Table 2) implies that these variables are mainly responsible for the seasonal variation observed in $A$ in the irrigation and fertigation cultivation systems. The high VPD can explain the reductions in $g_s$ in coffee trees of the irrigation system in the dry season due to the high temperature and low relative humidity in the region. This sensitivity of genotypes to VPD, associated or not with HD, is worrying and will probably be a critical factor for its cultivation in Rondônia, since climate projections indicate an increase in VPD in the Amazon in the decades to come [33]. These results corroborate the findings from a recent study carried out with

*C. arabica*, which demonstrates that VPD is a strong environmental indicator of global coffee productivity [35].

The observed values of *A* were relatively high in S2 and S3, considering the pattern normally presented by the species [5]. The seasonality of *A* can explain, at least in part, the seasonality of the growth rate of coffee observed in the region [60]. Fruiting may have contributed to increasing *A* in these seasons [21,23,24]. However, the increase in *A* in the coffee tree fruiting stage is also associated with the increase in $g_s$.

The variation observed in the anthocyanin index reveals a less-studied physiological aspect in coffee (*C. arabica* and *C. canephora*). Usually, high anthocyanin accumulation occurs in trees subjected to biotic or abiotic stress conditions [61]. Protection against radiation, especially UV radiation, is considered the main function of anthocyanins [62]. They act as a filter, preventing an excessive and harmful amount of energy from reaching the light-harvesting complexes of the photosystems [62]. The anthocyanins also are associated with the elimination of ROS resulting from photooxidation [61], and their synthesis can impede sugar-promoted feedback regulation of photosynthesis in leaves [62]. In this study, the highest anthocyanin content was observed in S1, the peak of the dry season. Interestingly, the increase in the anthocyanin index occurred in parallel with a decrease in the chlorophyll index. The change in the biochemical profile of the leaves may be due to the sensitivity of the plant metabolome to seasonal variation, as has been reported for other species [63].

In relation to the physiological variables, few differences were observed in the behavior of the genotypes, in the same system and evaluation period. This may be associated with the common genetic origin of the materials. Therefore, similarities are not restricted only to high yields [38,47].

The lower average growth rate of plagiotropic branches in coffee trees cultivated under dryland farming, during the annual cycle evaluated, reduced their productive potential for the next harvest because there were fewer nodes available to form reproductive buds [64]. However, a higher average rate of vegetative growth was expected in the dryland farming system because the low fruit yield (Table 3) would result in greater availability of photoassimilates to promote vegetative growth [25,64], especially between October and May. As the behavior of the photosynthetic traits was very similar, especially in S2, S3, and S4 (Figure 2), it is possible that a large part of *C* was allocated to the root system [64,65]. In addition, the total fixation of *C* by the tree, which is a function of *A* and the total leaf area, may have varied between treatments. The leaf area of a tree depends on the area of each leaf and the number of leaves on the tree. Both parameters were affected by the HD [5,16]. Thus, trees with smaller branches, fewer leaves, and/or smaller leaves have fewer available photoassimilates to sustain growth. The high temporal dependence of branch growth (Figure 6) corroborates this hypothesis. The more pronounced defoliation in coffee trees under dryland farming during the dry season (S1) may also have contributed to a possible reduction in *A* at the plant level [16,65].

In general, the yield obtained in the 2019/2020 harvest was low, considering the history of the experiment described previously (in the Experimental area and coffee cultivation section) and the potential of the genotypes [32,47]. Although there may be an effect resulting from the biennial production, it is not very expressive under our cultivation system [66]. The variable canonical 1 (Can. 1), constructed from the MANOVA for physiological variables (view Figure 2), and the bean yield exhibited a different behavior under the treatments (cultivation systems and clones) (Table 3); therefore, other physiological processes not evaluated in this study could have a high influence on coffee yield. Our results suggest that the range of optimal conditions for production, which is variable for each phase of the cycle, in some cases, was narrower than that often recorded for the photosynthetic process, as previously suggested [4]. The magnitude of the effect of climatic variables on each physiological process and yield depends on the cultivation practices, plant genotype, season, light intensity, and duration of stress [3,26]. For example, the final stages of flowering and the beginning of the fruiting (BBCH 59–70 [48]) coincided with high temperatures (view Figure 1) between late August and early October. The recorded values in this study were

not optimum and were thus harmful to the yield of *C. canephora* when they occurred in the phenological stages in which the trees were found [4]. Thus, the low yield obtained in the 2019/2020 harvest evaluated in this study can be explained largely by climate. Although localized irrigation adopted in irrigation and fertigation systems mitigates the effects of HD, it does not produce significant effects on VPD and Te.

The physiological traits evaluated, including branch growth rate, were similar between the fertigation and irrigation systems. However, fertigation had a positive effect on the yield of clone "08" and, in particular of "03", corroborating other studies [29]. On the other hand, the yield of clone "25" requires additional analysis to determine the potential benefits of fertigation [30]. However, the yield from clone "25" requires additional analysis to determine the potential benefits of fertigation. Complementary studies are needed to understand the effects of fertigation on the metabolism of coffee trees in Amazonian conditions and determine how the responses can be affected by climate variables, management, and genotypes. From an agronomic perspective, the technical/operational and economic feasibility of the practice must also be considered for recommending it. The high productive potential of clone "03" is already widely known by producers and technicians in the region [32,47]. Our results show that this genotype has mechanisms that allow it to adapt to adverse weather conditions, with or without a water deficit, at some critical stages of the production process. Its flowering, which is better distributed over successive blooms, may have reduced the proportion of reproductive structures exposed to the harmful temperatures of September.

Coffee production in the Brazilian Amazon is an important economic activity, which represents the main source of income for thousands of small farmers. Predominantly carried out in small areas, it occupies only about 85 thousand hectares, out of a total of approximately 1.6 million hectares deforested in the region. Despite a new cycle of expansion in the Amazon, coffee farming currently exerts little pressure on areas of native forest, since coffee plantations are predominantly located in deforested areas occupied with pastures. However, the use of irrigation on coffee plantations has had a growing and worrying impact on the region's water resources [67]. This situation is even more alarming considering that, as shown in the present study, some of the most economically relevant *C. canephora* genotypes for the region are vulnerable to drought and their production is significantly affected by water restriction. If climate change predictions are confirmed [68], the increases in atmospheric temperatures and changes in rainfall patterns can lead to more frequent and intense drought events that can considerably reduce water availability for irrigation, thus leading to significant reductions in coffee production on numerous properties. Importantly, some of these extreme weather events have already been experienced on a smaller scale in the last decade in many microregions of the state of Rondônia, which are already facing reductions in water availability for irrigation.

In summary, our findings highlight the need for new studies aimed at detailing the specificities of Amazonian Robustas genotypes ecophysiology. We suggest new research to assess the sensitivity and susceptibility to increasing VPD, with or without water restriction, especially with recent studies indicating a strong negative effect of this factor on coffee yield at relatively low thresholds [35]. It is also important to assess the hydraulic conductivity of different genotypes and to elucidate the mechanisms that enhance or limit it. In addition, studies should investigate the role of anthocyanins in the metabolism of coffee trees. Finally, it is still necessary to investigate the impact of climatic variables (such as temperature and VPD) on the plants' yield. This will help support the adoption of strategies aimed at sustainable cropping coffee in the Amazon.

## 5. Conclusions

For the three studied genotypes of *C. canephora* (clones "25", "08", and "03"), regardless of the cultivation system, there was seasonal variation in physiological variables, especially those related to gas exchange. Lower values were observed in the net assimilation rate of $CO_2$ ($A$) associated with lower stomatal conductance ($g_s$) in the dry season (S1). On the

other hand, in S1 higher values in the indices of anthocyanins in the leaves of the coffee trees were observed.

Amazonian Robustas genotypes cultivated in irrigated systems (irrigated and fertigated cropping systems) presented greater *A* and better indicators of chlorophyll *a* fluorescence in S1. The clone "03" presented worse indicators of chlorophyll *a* fluorescence in dryland system in S1.

The coffee yield of the three genotypes was lower in the dryland system. However, clone "03" had a much higher yield than clone "25" and clone "08". The productivity of the coffee plants cannot be explained by the physiological variables analyzed at the leaf level.

Our results also indicate that VPD has a strong effect on the metabolism of Robustas Amazonicos, directly affecting *A*. Therefore, at times of high demand for photoassimilates (such as flowering and fruiting), the occurrence of VPD above relatively low limits can lead to significant losses in the yield of Robustas Amazonicos. In a critical climate change scenario with forecasts of drier and warmer weather, the search for alternatives that reduce the effect of the VPD on coffee crops in the Amazon is urgent, at the risk of making the activity in the region unfeasible. In this sense, we recommend research aimed at the selection and/or development of genotypes that are less sensitive to VPD and the evaluation of shaded cropping systems or agroforestry.

**Author Contributions:** A.M.C.: Conceptualization, Methodology, Investigation, Data Curation, Writing—Original Draft, Project administration. P.E.d.M.S.: Writing—Review & Editing. T.R.d.S.: Investigation, Data Curation. L.L.L.: Conceptualization, Methodology, Investigation; R.G.A.: Conceptualization, Investigation, Writing—Original Draft. F.H.d.L.e.S.: Writing—Original Draft; M.C.E.: Writing—Original Draft. A.R.d.S.: Formal analysis. J.R.M.D.: Supervision, Project administration, Funding acquisition. F.G.S.: Supervision, Project administration, Funding acquisition. All authors have read and agreed to the published version of the manuscript.

**Funding:** This work was supported by the Research Support Foundation for the Development of Scientific and Technological Actions and Research of the State of Rondônia (*Fundação Rondônia de Amparo ao Desenvolvimento das Ações Científicas e Tecnológicas e à Pesquisa do Estado de Rondônia*-FAPERO) [Public Notice PAP AGRITECH n° 011/2018], Federal Institute of Rondônia (*Instituto Federal de Rondônia—IFRO*) [Process SEI 23243.007870/2018-90] and Federal Institute Goiano (Instituto Federal Goiano—IF Goiano) [translation, processing and publication of this article].

**Institutional Review Board Statement:** Not applicable.

**Informed Consent Statement:** Not applicable.

**Data Availability Statement:** Not applicable.

**Acknowledgments:** The authors would like to thank the Federal Institute Goiano (IF Goiano), the Federal Institute of Rondônia (IFRO), and the Federal University of Rondônia (UNIR) for the support and resources provided available to carry out this study. We also thank the managers and other employees of the respective institutions, and the National Council for Scientific and Technological Development—CNPq [grants: 309733/2021-9].

**Conflicts of Interest:** The authors declare that they have no known competing financial interest or personal relationships that could have appeared to influence the work reported in this paper.

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
