# Peer review of "Seasonal Variation in Physiological Traits of Amazonian Coffea canephora Genotypes in Cultivation Systems with Contrasting Water Availability"

_agronomy, doi:10.3390/agronomy12123197_

Round 1

Reviewer 1 Report (New Reviewer)

The authors report a study on the physiological response of Coffea canefora cultures (three genotypes) grown in the Brazilian Amazon.

As the authors report, C. canephora is grown mainly in Rondonia, few studies have been done on other areas of Brazil. The authors evaluate photosynthetic pigments, photosynthetic yield, transpiration processes, water content in leaves and correlate these parameters with seasonal variations. In particular, in these tropical areas there are two main seasons (a dry season and a rainy season) with different rainfall and light radiation.

The authors demonstrated how the alternation of the two seasons influenced the levels of chlorophyll pigments, leaf anthocyanins, CO2 exchanges.

The manuscript is clear, and straightforward. The authors argue their thesis well and abundantly discuss the results obtained. The theme of seasonal variations is common to many wild and agronomic crops. For this reason, I suggest the authors to integrate the manuscript references with this recent work:

Pucci, M.; et al. Different Seasonal Collections of Ficus carica L. Leaves Diversely Modulate Lipid Metabolism and Adipogenesis in 3T3-L1 Adipocytes. Nutrients 2022, 14, 2833. doi: 10.3390/nu14142833 

The manuscript needs few corrections, mostly typos.

Author Response

Dear reviewer, we thank you for your contributions and considerations. We arrived at an improved version of the manuscript after following your suggestions.
We list below.

Point 1: The manuscript is clear, and straightforward. The authors argue their thesis well and abundantly discuss the results obtained. The theme of seasonal variations is common to many wild and agronomic crops. For this reason, I suggest the authors to integrate the manuscript references with this recent work:

Pucci, M.; et al. Different Seasonal Collections of Ficus carica L. Leaves Diversely Modulate Lipid Metabolism and Adipogenesis in 3T3-L1 Adipocytes. Nutrients 2022, 14, 2833. doi: 10.3390/nu14142833.

The manuscript needs few corrections, mostly typos.

Response 1: We thank the reviewer for their contributions. We did a general review of the text to correct any typos or even writing errors. We also accept the recent bibliography suggested.

Reviewer 2 Report (New Reviewer)

Introduction: The different topics are well documented. Arguments to support the objectives are clear and references seems enough to clarify the interest on the physiologic aspects related with the use of water and adaptation behavior of Conilons to the Amazonian region. 15% of Brazilian production in Amazonian region is a strong reason to justify this study.

It would be important to add some information about the use and impact of the fertigation systems in coffee (what we know about that? What would be the impact of this practice on some variables like yield or plant growth?)

 Material and Methods:

Line 157: Please include a more precise information about the three genotypes (clones). Even if there are precedent references, it is not clear what are the specific differences among them in terms of genetics (yield potential, adaptability, quality) and origin. This information will allow to better understand the results of performance regarding the different experimental situations.

 Line 239: Regarding the metrics, the use of MT (metric tons) would be more comprehensible for lectors than Mg (megagrams). Please consider.

 Results and discussion:

Line 280: Table 1. Please check the values in the table. Min and Max values for RH% do not correspond to the column. Please correct them.

 Line 484: The importance if VPD is crucial for the future adaptation of coffee to particular regions as Amazonian zone. The results of this work have been confirmed also by a quite recent study on Arabica coffee showing that VPD is a strong physiologic indicator of global coffee productivity (Kath J et al, Nature Food, October 2022). Please check and include this key reference.

 Line 528: Besides the effect of all the factors mentioned as responsible of the low yield observed in the evaluation period, authors must also consider/discuss the “biennial effect on yield” which is a normal condition observed in coffee production. Please consider.

 Conclusions:

General conclusions are according to the main results and observations. They provide some interesting answers to the original experimental questions.  

Given the recent evidence (paper mentioned above), I believe that it could be important to mention the importance of VPD to explain the risk of yield reduction in coffee (Robusta but also Arabica) regarding the exposure to specific climatic factors. Please consider.

A general description of the genetic origin of the winner clone 03 would be useful to understand their potential in the Amazonian region and the possibility to include new varieties with the same genetic background. Please include this information and discus on it.  

Author Response

Dear reviewer, we thank you for your contributions and considerations. We arrived at an improved version of the manuscript after following your suggestions.
We list below.

Point 1: Introduction: The different topics are well documented. Arguments to support the objectives are clear and references seems enough to clarify the interest on the physiologic aspects related with the use of water and adaptation behavior of Conilons to the Amazonian region. 15% of Brazilian production in Amazonian region is a strong reason to justify this study.

It would be important to add some information about the use and impact of the fertigation systems in coffee (what we know about that? What would be the impact of this practice on some variables like yield or plant growth?)

Response 1: We thank the reviewer for contributions and suggestions. We have included in the introduction a brief quote about potential benefits of fertigation for coffee cultivation. Although, in the literature, there is a scarcity of studies directly evaluating the impact of fertigation on coffee trees in relation to conventional fertilization system in irrigated crops.

Point 2: Material and Methods:

Line 157: Please include a more precise information about the three genotypes (clones). Even if there are precedent references, it is not clear what are the specific differences among them in terms of genetics (yield potential, adaptability, quality) and origin. This information will allow to better understand the results of performance regarding the different experimental situations.

Response 2: Thanks to the reviewer for the suggestion. We accepted the suggestion and described the genotypes in little more detail. However, it is important to emphasize that so far there are no studies addressing genetic and/or (eco)physiological differences in genotypes. Our study will be the first with Amazonian Robusta, about ecophysiological aspects.

Point 3: Line 239: Regarding the metrics, the use of MT (metric tons) would be more comprehensible for lectors than Mg (megagrams). Please consider.

Response 3: We appreciate and accept the reviewer's suggestion.

Point 4: Results and Discussion:

Line 280: Table 1. Please check the values in the table. Min and Max values for RH% do not correspond to the column. Please correct them.

Response 4: We thank the reviewer for the observation. And we made a small change to the table to improve the presentation of the data and its understanding. The values in the column "Maximum" referred to by the reviewer refer to temperature and not to air humidity. We believe that the table became clearer after reorganizing the columns following an ordered pattern (minimum, average, maximum - when applicable).

Point 5: Line 484: The importance if VPD is crucial for the future adaptation of coffee to particular regions as Amazonian zone. The results of this work have been confirmed also by a quite recent study on Arabica coffee showing that VPD is a strong physiologic indicator of global coffee productivity (Kath J et al, Nature Food, October 2022). Please check and include this key reference.

Response 5: We thank the reviewer for the suggestion and indication of the bibliography. Very pertinent and corroborates the findings of our work, even though it was dedicated to arabica coffee. We reinforce in the manuscript this coherence between the two studies.

Point 6: Line 528: Besides the effect of all the factors mentioned as responsible of the low yield observed in the evaluation period, authors must also consider/discuss the “biennial effect on yield” which is a normal condition observed in coffee production. Please consider.

Response 6: Thanks again to the reviewer for the suggestion and contribution. We accept the suggestion and point out in the text that the low productivity may have been due to the biennial production, although it is a less expressive factor in our cultivation systems, compared to other regions, mainly with C. arabica. This is due to genetics, environmental/climatic conditions and cultural practices - mainly pruning.

Point 7: Conclusions:

General conclusions are according to the main results and observations. They provide some interesting answers to the original experimental questions. 

Given the recent evidence (paper mentioned above), I believe that it could be important to mention the importance of VPD to explain the risk of yield reduction in coffee (Robusta but also Arabica) regarding the exposure to specific climatic factors. Please consider.

Response 7: We appreciate and accept the reviewer's suggestion. We include a brief paragraph in the conclusion focusing on the VPD.

Point 8: A general description of the genetic origin of the winner clone 03 would be useful to understand their potential in the Amazonian region and the possibility to include new varieties with the same genetic background.

Response 8: Thanks to the reviewer for this further suggestion. We found it more convenient to make this description in the materials and methods, as suggested by the reviewer, and briefly in the discussion. The lack of specific studies that detail and better describe the genotype in terms of metabolome, physiology, morphoanatomy, prevents us from reaching more precise and reliable conclusions to explain why this genotype was superior.

This manuscript is a resubmission of an earlier submission. The following is a list of the peer review reports and author responses from that submission.

Round 1

Reviewer 1 Report

The Manuscript evaluated the physiological parameters and yield of Coffea canephora at four different periods in three cultivation systems with three genotypes. The results can well explain the impact of seasonal climatic changes on the photosynthesis and yield of Coffea canephora in the Brazilian Amazon. 

There are still some problems that need to be revised as following.

  1. Figure serial numbers and indication in the context are chaos and readers will get confused when looking for relevant figures.
  2. Figure 3 (in the original manuscript) were showed in several place or these figures had the similar meaning. It is suggested to combine these figures together.
  3. One of the purposes of crop-environment research is how to achieve stable crop production. A discussion therefore, is recommended on how to adapt Coffea canephora production to weather conditions and how to maintain high yields under adverse weather conditions in the Brazilian Amazon.

Reviewer 2 Report

Generally speaking, the topic is of interest, which is helpful to understand the response mechanism of eco-physiology of differrent coffee genotypes to the climate variations in the Brazilian Amazon.

My general opinions to this manuscript is that the new scientific findings is not well summarized, I have this opinion based on the (at least) two shortcomings in this manuscript:

First, the experiment design in this manuscript is a matrix of {three cultivation systems [irrigation (I), fertigation (F), and dryland farming (D)] Х three genotypes [clones “25” (G1), “08” (G2), and “03” (G3)] Х four seasons (S1, S2, S3, and S4)}, while the role of cultivation systems is weak in the discussion of the new findings in this manuscript. For example, in the Abstract, just mentioned '...even under irrigated conditions (Line 25)', which means the other two cultivation systems are not worthy mentioning? In the section '5. Conclusions', just to say '... in all cultivation systems (Line 539)', it is too general.

The second point is that in the Abstract Line 25-27, 'The physiological variables analyzed at the leaf level, even in different periods, did not explain the differences in the yield of C. canephora.' is quite a confused description, if the physiological variables cannot explain the yield, which factors can explain?

Specific comments:

Line 22-23, the description of 'As expected, the values of gs, E, and A were lower in S1, while the values of VPD Leaf-ar , T Leaf, and IWUE were higher.' is quite general, could the authors give more qualitative (for example: a bit lower, obviously lower, or slightly higher, significantly higher) evaluation results? Better to present the quantitative evaluation results;

Line 24, the description '...climatic seasonality affects the photosynthesis...', we can say it is a common knowledge, without the research in this manuscript, we can also get the conclusion like this, it is strongly suggested to specify;

Why there are two 'Figures 1'? Besides, for the description of '...during the experimental period' (Line 108 and Line 112), it is suggested to clearly indicate the exact timeline;

It seems that Figure 3 is repeated a lot of times in this manuscript (Line 258-259, Line 264-265, Page 7; Line 267Page 7-Line 268 Page 8; Line 278-279, Page 8; Line 284-285, Line 289-290, Page 9; Line 397-398, Page 15; Line 415-416, Page 16; Line 479-480, Page 17), so please check whether it is an editing mistake?

Following up the three figures (Line 105-118), there should be a summarizing description on the climatic features during the experiment period; besides, these three figures (Line 105-118) should be put in the section '3. Results'.

Reviewer 3 Report

Seasonal variation in physiological parameters and yield of Coffea canephora in the Brazilian Amazon

By

Custodio A.M. et al.

General Remarks

The present study examines the seasonal variation in physiological traits mainly related to photosynthetic performance (gas exchange, Chla fluorescence, pigments) and its probable relationship with yield of 3 coffee genotypes under 3 different cultivation systems.  It is evident that the authors have devoted a significant amount of time and effort both for collecting the data as well as preparing the manuscript, however in my opinion the aim of the study is a bit confusing and leads to a discussion that is highly speculative about the physiological processes that could explain the observed differences. Although the authors clearly mention the significant role of leaf traits to total C fixation efficiency and consequently investments to biomass production and yield, they have not examined any of them (e.g. at least leaf area or leaf mass per area) either as component of the genotypic variation or the phenotypic plasticity of each genotype. It is expected that structural acclimation processes under different environmental factors (temperature, water regime etc) not only lead to alterations of leaf area and number (as mentioned for HD) but they also include a great number of leaf plastic responses from the subcellular to the organ level affecting all aspects of photosynthetic performance (e.g. stomatal and mesophyll conductance, photochemical efficiency etc). Differences in biological cycle have also not been taken into account. Physiological differences are expected between vegetative and reproductive stages as well as post-anthesis and bean filling.

I believe that the nature of the present study with a combination of treatments, genotypes, environmental factors etc cannot provide an in depth examination of the mechanisms and therefore authors should avoid the interpretation of their results in a way that leads to a high speculative discussion about the physiological processes. It should be noted that correlation does not imply causation.  

Authors should reconsider their approach for presenting their results. I think the main focus should be given on the differences of the responses among genotypes and treatments and physiological traits should be used as indicators of different responses while toning down the physiological interpretations. Please find my specific comments below.

Introduction

It is generally well written and the focus here is on the cultivation practices and production as well as on the genotypes mostly used. But authors should incorporate more background about what is already known about Coffea ecophysiology and its connection to yield e.g. Da Matta et al. 2007 (citation 1 used by the authors).

Line 34. “yield directly associated with the photosynthetic process”: I was a bit surprised by this sentence given that there is a great discussion about this relation. I do not think this is stated in the review the authors cite. The reference used highlights the significance of leaf and canopy traits.

Line 42-55 Perhaps it would be more appropriate to give some background on Coffea photosynthetic responses rather than such a general content e.g. what is known about stomatal responses in the species?

There is a great number of studies on the ecophysiology of the species given its significance.

Materials and Methods

Line 93-118 I think the presentation of the climatic data is very extensive. I think a table with the average value for S1-S4 would provide the necessary information in a more direct way especially given that physiological measurements are instantaneous and there is not enough information on the rest of the life cycle that justifies such a detailed presentation.

Minor correction: please check figures numbering

Lines 124-128: How is this information relevant?

Line 144 why were these specific genotypes selected? Please elaborate

Line 148-157: the fact that measurements were conducted at different stages should also be discussed

Line 159-164: so replication is 2 leaves(plants) x 4 replicates (Line 144)? Please clarify n.

Line 166-173: I am a bit confused. How were chlorophylls measured?

Line 169-170: Why were data for S2 obtained indirectly? Please explain

I acknowledge the fact that many studies use the values of chlorophyll-polyphenol meters without further conversion however if the authors aimed to identify meaningful physiological differences expression per leaf area/mass should have been considered.

Line 182-183: JIP test is an appropriate tool for field studies and in particular acclimation responses but needs to be combined with additional traits (please see e.g. Bussotti et al., 2010: The JIP test: a tool to screen the capacity of plant adaptation to climate change). Also, please check that this are the appropriate citations. Is 11 relative?

Line 201: is this the ambient irradiance for the specific leaf? How did the authors decide to measure at this light intensity?

I also really think that given that the measurements were not done in a controlled environment, there is no need to present all the variables measured by LICOR but the ones that may contribute to a meaningful discussion. If authors had performed A/Ci curves then it would make sense to refer to Ca, Ci etc. These are very sensitive measurements and I do not think that their presentation makes any sense when there are so many varying conditions.  

Line 205-212: Why RWC and not leaf water potential that is a better indicator of the condition of the plant? Later on in the text authors imply that they measured RWC at different time points. Is this why they used four segments? Please explain

Line 215-220 there are some abbreviations concerning vegetative growth and production mentioned in the results as p.ar1, o.r1 p.med and o.med that are not previously mentioned or explained. Please provide the necessary information

Line 225-226. They were tested but did the meet the criteria? Please rephrase.

Line 231: was multicollinearity taken into account? How meaningful is it to include all the traits given that many of them are not just highly correlated but many are also derivatives of the rest? The plots are not easy to read and the highly correlated traits create bias. Was values normalization considered?

In my opinion statistical analysis should be adapted to the suggested approach i.e. intra and intergenotypic variation of seasonal responses and treatment effect.

Trait correlation networks are a powerful tool and innovative but it is rather obvious that it just shows the obvious and expected correlations of traits/parameters that are auto correlated due to the basic principles followed by the method of their measurement.

Results

I am pretty sure this is not the authors’ s fault, but in my pdf version Figure 3 appears in the text 6 times!

Given the number of traits I think it would be very helpful if the legend included the abbreviations.

Also, why use all Chla fluorescence parameters? For the aims of this study PIAB is a gross indicator of stress. What is the purpose of presenting all these parameters if you did not design a study that examines photochemistry in depth?

So what are Can. 1 and Can.2 representing? In every CDA these are two new variables that need some interpretation.

Table 2: what are G1, G2 and G3?

Discussion

Discussion should change according to all the previous comments-suggestions.  According to the study design and the lack of a limitation analysis (stomatal, diffusional etc) there is no way to safely conclude whether the effect of all these environmental factors impacted only the biochemical phase of photosynthesis.

Of course as explained the majority of these responses were expected. These are not novel findings and many of the statements regarding the physiological mechanisms are not supported by the results, hence they are speculative. Perhaps authors should focus on what is the new information that their study can provide e.g. evaluating and comparing the responses of the three genotypes along the seasons and among the treatments.

Conclusions are quite trivial and should be changed according to previous suggestions.

Reviewer 4 Report

The authors have designed an experiment under uncontrolled conditions. It is demonstrated because they want to study genotypes, fertigation, irrigation and dryland farming (four factors) (page 2) in their objectives. But, the experimental design (page 4) only contemplates a split plot design with 2 factors (genotypes and crop season). There is no correspondence between the objectives of the work and the experimental design. If the statistical design is a split plot, a correct analysis should be performed for this design. The authors do not present it in their manuscript. You can consult the references for this analysis.

In addition, the authors have not found an explanation for the yield according to the physiological parameters studied. There are many physiological variables and their study seems to be a result of all of them without seeking an explanation for a working hypothesis. All this is supported by the fact that in the conclusions the authors do not consider yield to be important. In addition, not only the yield is important but also the quality.

For all these reasons, the authors should reorient the design to have fewer degrees of freedom and more controlled conditions, modifying the factors that are intended to be studied only; and carry out an appropriate statistical analysis for the split plot design.